# Pressurized Intraperitoneal Aerosol Chemotherapy (PIPAC) for Gastric Cancer Peritoneal Metastases: Results from the Lithuanian PIPAC Program

**DOI:** 10.3390/cancers16172992

**Published:** 2024-08-28

**Authors:** Martynas Luksta, Augustinas Bausys, Neda Gendvilaite, Klaudija Bickaite, Rokas Rackauskas, Marius Paskonis, Raminta Luksaite-Lukste, Anastasija Ranceva, Rokas Stulpinas, Birute Brasiuniene, Edita Baltruskeviciene, Nadezda Lachej, Juste Bausiene, Tomas Poskus, Rimantas Bausys, Skaiste Tulyte, Kestutis Strupas

**Affiliations:** 1Clinic of Gastroenterology, Nephrourology, and Surgery, Institute of Clinical Medicine, Faculty of Medicine, Vilnius University, 03101 Vilnius, Lithuania; 2Department of Abdominal Surgery and Oncology, National Cancer Institute, 08406 Vilnius, Lithuania; 3Laboratory of Experimental Surgery and Oncology, Translational Health Research Institute, Faculty of Medicine, Vilnius University, 03101 Vilnius, Lithuania; 4Department of Radiology, Nuclear Medicine and Medical Physics, Institute of Biomedical Sciences, Faculty of Medicine, Vilnius University, 03101 Vilnius, Lithuania; 5Hematology, Oncology, and Transfusion Medicine Center, Vilnius University Hospital Santaros Klinikos, 08410 Vilnius, Lithuania; 6National Center of Pathology, Affiliate of Vilnius University Hospital Santaros Klinikos, 08406 Vilnius, Lithuania; 7Department of Medical Oncology, National Cancer Institute, 08406 Vilnius, Lithuania; 8Institute of Clinical Medicine, Faculty of Medicine, Vilnius University, 03101 Vilnius, Lithuania

**Keywords:** gastric cancer, pipac, peritoneal metastasis

## Abstract

**Simple Summary:**

Peritoneal metastases from gastric cancer are linked to a poor prognosis, with median survival ranging from 2 to 9 months. Standard treatments, including systemic chemotherapy and targeted therapies, have demonstrated only limited effectiveness. Pressurized intraperitoneal aerosol chemotherapy (PIPAC) is an experimental approach under investigation for treating these metastases, but its widespread clinical use is hindered by insufficient evidence regarding its safety and efficacy. This retrospective study presents outcomes from the first PIPAC program in Lithuania, where 32 patients underwent 71 PIPAC procedures between 2015 and 2022. Intraoperative and postoperative complications occurred in 4.2% of cases. Although reductions in peritoneal carcinomatosis index (PCI) and ascites volume were noted, they were not statistically significant. The median overall survival after PM diagnosis was 12.5 months. These findings suggest that PIPAC is a safe and feasible treatment, but further research is needed to establish its efficacy.

**Abstract:**

Background: Peritoneal metastases (PM) of gastric cancer (GC) are considered a terminal condition, with reported median survival ranging from 2 to 9 months. Standard treatment typically involves systemic chemotherapy alone or combined with targeted therapy or immunotherapy, though efficacy is limited. Pressurized intraperitoneal aerosol chemotherapy (PIPAC) has emerged as a novel technique for treating GC PM, although it remains an experimental treatment under investigation. This study aimed to summarize the outcomes of GC PM treatment with PIPAC from the Lithuanian PIPAC program. Methods: All patients who underwent PIPAC for GC PM at Vilnius University Hospital Santaros Klinikos between 2015 and 2022 were included in this retrospective study. The safety of PIPAC was assessed by postoperative complications according to the Clavien–Dindo classification. Efficacy was evaluated based on the peritoneal carcinomatosis index (PCI), ascites dynamics throughout the treatment, and long-term outcomes. Results: In total, 32 patients underwent 71 PIPAC procedures. Intraoperative and postoperative morbidity related to PIPAC occurred after three (4.2%) procedures. Following PIPAC, there was a tendency towards a decrease in median PCI from 10 (Q1 3; Q3 13) to 7 (Q1 2; Q3 12), *p* = 0.75, and a decrease in median ascites volume from 1300 mL (Q1 500; Q3 3600) at the first PIPAC to 700 mL (Q1 250; Q3 4750) at the last PIPAC, *p* = 0.56; however, these differences were not statistically significant. The median overall survival after PM diagnosis was 12.5 months (95% CI 10–17), and the median survival after the first PIPAC procedure was 5 months (95% CI 4–10). Conclusions: PIPAC is a safe and feasible treatment option for GC PM; however, well-designed prospective studies are needed to fully assess its efficacy.

## 1. Background

Gastric cancer (GC) ranks fifth globally in incidence and fourth in mortality, affecting over one million patients annually [1]. The disease often presents a significant challenge due to the lack of effective screening programs and the frequently asymptomatic nature of its early stages, leading to diagnoses at more advanced stages [2,3]. At these stages, up to 30% of patients may present with peritoneal metastases (PM) [4]. Moreover, a significant proportion of patients develop metachronous PM despite previous radical treatment for GC [5]. PM in gastric cancer is generally considered a terminal condition, with median survival rates reported to range from 2 to 9 months [6]. The standard treatment approach for GC PM typically includes systemic chemotherapy, which may be used alone or in combination with targeted therapy or immunotherapy; however, the efficacy of these treatments is often limited [7]. A response rate of less than 14% to systemic treatment can be expected for patients with PM, compared to a response rate of approximately 40% for patients with liver, lung, or bone metastases [8]. One of the primary challenges in managing PM is overcoming the plasma–peritoneal barrier. This barrier significantly impedes the effective delivery of chemotherapeutic agents directly to the peritoneal cavity. To address this issue, various innovative approaches, including the use of nanoparticles for drug delivery [9] and intraperitoneal chemotherapy [10], have been proposed and are currently at different stages of development. One of the primary advantages of intraperitoneal chemotherapy is the minimized systemic absorption of anticancer drugs administered into the peritoneal cavity. This results in higher regional concentrations of the drugs and extended direct exposure time to PM and free cancer cells [11]. Different methods of drug delivery are utilized: normothermic intraperitoneal chemotherapy via intraperitoneal port systems is more commonly used in Asian countries, while hyperthermic intraperitoneal chemotherapy (HIPEC) is preferred in the Western world [11]. The combination of hyperthermia and chemotherapy appears to be beneficial for three main reasons: (I) hyperthermia itself has a selective cytotoxic effect on cancer cells; (II) hyperthermia enhances tissue perfusion and oxygenation, potentially increasing the penetration of cytotoxic drugs; (III) several chemotherapeutic compounds, particularly platinum derivatives, exhibit enhanced cytotoxicity when used in conjunction with hyperthermia [11]. HIPEC may be applied solely as a neoadjuvant treatment [12], but it is usually combined with complete or near-complete cytoreductive surgery. However, for this invasive treatment, patients need to be in good condition, which is often compromised in advanced GC. While the benefits of CRS and HIPEC have been demonstrated in several selected patient cohorts, their efficacy generally remains highly controversial [8]. Pressurized intraperitoneal aerosol chemotherapy (PIPAC) represents a novel approach in this domain aimed at improving drug distribution within the peritoneal cavity [6]. PIPAC is a minimally invasive technique that utilizes physical principles to optimize the delivery of chemotherapeutic drugs. Key mechanisms of PIPAC include (I) optimizing drug dispersion through aerosol delivery, (II) enhancing drug penetration by increasing intraperitoneal hydrostatic pressure, (III) minimizing blood outflow during drug application, and (IV) controlling environmental parameters within the peritoneal cavity to target tissues more effectively [8]. Additionally, PIPAC allows for repeated drug applications and provides the ability to assess tumor responses objectively by comparing biopsies obtained at different stages of treatment [8].

The first PIPAC procedure was performed in 2011 in Germany [9], and the first and only PIPAC program in the Baltic region was established in 2015 at Vilnius University Hospital Santaros Klinikos [13]. Since then, numerous studies have examined the application of PIPAC for GC PM, as summarized in a recent systematic review [14]. These studies suggest that PIPAC holds promise as a treatment option. Nevertheless, conclusive evidence on its effectiveness remains limited, underscoring the need for further investigation. Thus, this study aims to evaluate outcomes following GC PM treatment with PIPAC at the first Baltic center.

## 2. Materials and Methods

### 2.1. Study Design and Ethics

This retrospective cohort study was carried out at Vilnius University Hospital Santaros Klinikos. Prior to its commencement, approval was obtained from the Vilnius Regional Biomedical Research Ethics Committee (No. 2020/11-1279-761), with a waiver of informed consent granted by the authority. The study adhered to the principles outlined in the Declaration of Helsinki.

### 2.2. Patients

All patients who underwent PIPAC for GC PM between 2015 and 2022 at Vilnius University Hospital Santaros Klinikos were included in the study. Patient data, encompassing demographic details and clinicopathologic features like gender, age, previous cancer treatment history, and the Peritoneal Carcinomatosis Index (PCI) score for each PIPAC procedure, were obtained from the institutional electronic database. Additionally, comprehensive treatment-related variables were recorded to provide a detailed overview of the clinical and surgical aspects of the procedures. These variables included the duration of surgery, intraoperative blood loss, and the specific chemotherapeutic agents used during PIPAC. Postoperative complications were documented and graded according to the Clavien–Dindo score, ensuring a standardized assessment of surgical outcomes. Furthermore, data on other oncological treatments administered alongside PIPAC were collected.

### 2.3. PIPAC Procedure

Our center’s protocol for the PIPAC procedure has been previously published [13]. In summary, potentially eligible patients who expressed willingness to undergo experimental treatment with PIPAC were evaluated by a multidisciplinary team, and treatment decisions were made on an individual basis for each case.

All surgeries were conducted under general anesthesia, with a single 2.0 g dose of cefazolin administered intravenously for antibiotic prophylaxis before the incision. At the start of the surgery, two balloon trocars were inserted into the abdominal wall after establishing a 12 mmHg capnoperitoneum via open access. The Peritoneal Carcinomatosis Index (PCI) was then assessed, and biopsies were taken from four different regions of the peritoneal cavity. Ascitic fluid was completely drained, and its volume was recorded and sent for cytological analysis. Then, PIPAC was utilized using a 9 mm Capnopen (CapnoPharm, Tübingen, Germany) microinjection pump connected to an intravenous high-pressure injector. A solution of 150 milliliters of isotonic NaCl 0.9% containing cisplatin at a dose of 7.5 mg/m^2^ body surface area and doxorubicin at 1.5 mg/m^2^ body surface area was delivered through the microinjection pump at a pressure of 200 psi and a rate of 0.5 mL/s, creating an aerosol within the abdominal cavity. Intra-abdominal pressure was maintained at 12 mmHg throughout the 30 min procedure.

Subsequent PIPAC procedures were scheduled at 6-week intervals, allowing for ongoing assessment and treatment adjustments as needed. The decision to administer PIPAC bidirectionally, with systemic chemotherapy applied between PIPAC procedures or as a unimodal treatment option, was made on a case-by-case basis during multidisciplinary team meetings.

### 2.4. Study Outcomes

Study outcomes included (1) overall survival (OS), (2) postoperative morbidity, (3) PIPAC impact on Peritoneal Carcinomatosis Index (PCI), and (4) ascites volume.

Overall survival (OS) was defined in two ways: from the time of peritoneal metastasis (PM) diagnosis to death and from the first PIPAC procedure to death. Data on survival and date of death were collected from the Lithuanian National Cancer Registry.

Postoperative morbidity was assessed through complications arising within 30 days after the PIPAC procedures and was classified according to the Clavien–Dindo classification. The impact of PIPAC on the PCI score was evaluated to determine changes in the extent of peritoneal carcinomatosis following the treatment. Ascites volume was monitored by measuring the amount of fluid drained during each PIPAC procedure, providing insights into the treatment’s effect on ascites management.

### 2.5. Statistical Analysis

All statistical analyses were conducted using the statistical program R studio version 2022.12.0+353 (Integrated Development for R; RStudio, PBC, Boston, MA, USA). The normality of the data was tested by using the Shapiro–Wilk normality test. Continuous variables are presented as median with an interquartile range. Related samples were compared using the Wilcoxon signed-rank test. Overall survival rates were analyzed using the Kaplan–Meier method.

## 3. Results

### 3.1. Baseline Characteristics

In total, 32 patients underwent 71 PIPAC procedures. The distribution of the number of PIPAC procedures performed per patient was as follows: nine patients (28.1%) underwent one procedure, nine patients (28.1%) had two procedures, thirteen patients (40.6%) received three procedures, and one patient (3.1%) had five procedures.

The baseline clinicopathologic characteristics of these patients are detailed in Table 1. Among the cohort, 29 patients (90.6%) received PIPAC for synchronous peritoneal metastasis (PM), while 3 patients (9.4%) were treated for metachronous PM.

Prior to undergoing PIPAC, 29 patients (90.6%) had received systemic chemotherapy. The chemotherapy regimens varied, with patients being treated using different schemes: FOLFOX (n = 14), XELOX (n = 7), EOX (n = 5), FOLFIRI (n = 5), and FLOT (n = 2). Specifically, 18 patients (56.3%) had been administered first-line chemotherapy, 7 patients (21.9%) received second-line treatment, and 4 patients (12.5%) were given third-line chemotherapy prior to PIPAC. Additionally, one patient (3.1%) chose to forego systemic chemotherapy and instead received adjuvant radiotherapy following gastrectomy before initiating PIPAC treatment. Notably, two patients (6.3%) received PIPAC as their first-line treatment. Twenty patients (62.5%) received PIPAC as a bidirectional treatment in combination with systemic therapy, while twelve patients (37.5%) received it as a unimodal treatment.

### 3.2. PIPAC Procedure Characteristics

Following PIPAC, the median PCI decreased from 10 (Q1 3; Q3 13) to 7 (Q1 2; Q3 12), *p* = 0.75 (Figure 1). At baseline, 11 patients (34.4%) had ascites. Following PIPAC, the median volume of ascites decreased from 1300 mL (Q1 500; Q3 3600) at 1st PIPAC to 700 mL (Q1 250; Q3 4750) at last PIPAC, however, the difference was not significant, *p* = 0.56. There was no need for conversion to open surgery throughout the study. In terms of morbidity, intraoperative and postoperative complications related to PIPAC occurred in 3 out of 71 procedures (4.2%). Among the patients, two experienced postoperative complications: one patient (1.4%) developed severe postoperative neutropenia (Clavien–Dindo score 2), and another (1.4%) developed an intra-abdominal abscess that required ultrasound-guided drainage (Clavien–Dindo score 3a). Additionally, one patient (1.4%) suffered an intraoperative bowel perforation during the initial port placement due to severe adhesions. However, this injury was repaired during the surgery, and the postoperative course was uneventful.

### 3.3. Long Term Outcomes

The median time to follow-up after diagnosis of PM was 13 (Q1 9; Q3 18) months. The median OS after the PM diagnosis by Kaplan–Meier analysis was 12.5 (95% Cl 10–17) months (Figure 2), and after the first PIPAC procedure, it was 5 (95% Cl 4–10) months (Figure 3).

## 4. Discussion

This study presents the results of the first PIPAC for the GC PM program in the Baltic region. Our findings indicate that PIPAC is a safe procedure with a postoperative complication rate of less than 5% without complications threatening life. Also, we found a tendency that PIPAC may reduce the PCI and ascites volume, although these results failed to show statistical significance. Present findings contribute to the growing body of evidence supporting PIPAC as a safe, feasible, and potentially beneficial treatment option for patients with GC PM. The study provides valuable insights into the effectiveness of PIPAC in managing this challenging condition and highlights the need for further research to confirm its impact and optimize treatment protocols.

Many endpoints are utilized in existing studies to evaluate the efficacy of PIPAC. Objective assessment of therapy response is essential for evaluating new cancer treatments, but it presents difficulties for PM due to the limitations of current radiological imaging techniques, especially in patients with low-volume disease. Small peritoneal metastases are challenging to detect with imaging, and measuring changes in their volume is even more complex. Neither computed tomography nor magnetic resonance imaging reliably assesses tumor adherence or extensive involvement of the small bowel or mesentery. Consequently, peritoneal metastases are often categorized as “non-measurable disease” and excluded from response evaluations, which means patients with these metastases are frequently omitted from randomized studies [15]. Repeated laparoscopy used for PIPAC allows for direct monitoring of the efficacy of multiple cycles of intraperitoneal chemotherapy by measuring various parameters, including PCI dynamics, ascites volume, and histological response. In our cohort, we observed a trend towards a decrease in the median PCI from 10 to 7 and a reduction in ascites volume following PIPAC treatment, although these changes did not achieve statistical significance. These findings align with previous research indicating that PIPAC may help reduce ascites in patients with GC [2]. However, the evidence regarding PIPAC’s impact on PCI is less clear. Several studies have suggested that PIPAC may not significantly decrease PCI [2,13,16,17,18,19,20], and there is no definitive evidence linking a reduction in PCI with improved patient outcomes. Such PIPAC impact on PCI is not surprising, as peritoneal metastases may not completely disappear following intraperitoneal chemotherapy but instead become non-viable and fibrotic. Tumor regression grading scores are widely used in the neoadjuvant setting for primary tumors; for instance, the Becker grading system is commonly used for GC [21]. Thus, a similar peritoneal regression grading score (PRGS) has been proposed for objective intraperitoneal chemotherapy response assessment in PM [21]. This four-tier score considers acellular mucin and infarct-like necrosis as regression features [21], and it has prognostic value, as a better response is associated with increased survival rates [22]. Unfortunately, at the time our study was conducted, PRGS was not included in our standard histological reports. As a result, we were unable to explore its potential relationship with treatment efficacy and patient outcomes in our cohort. Overall, while our study supports the potential benefits of PIPAC, especially in managing ascites, further research incorporating PRGS and other relevant endpoints is needed to fully understand the impact of PIPAC on disease progression and patient survival.

Another important aspect of PIPAC treatment is its safety and feasibility. Our results showed acceptable safety of PIPAC, with intraoperative and postoperative morbidity occurring in only 4.2% of procedures and 0% mortality. These complications included one case of neutropenia, one intra-abdominal abscess, and one bowel perforation, all of which were managed effectively. Our results are consistent with previous studies showing that the rate of severe or life-threatening complications following PIPAC ranges between 0.7% and 25% [2,16,17,20,23,24,25,26]. Such a wide range of complication rates may arise from the use of different grading systems across the studies. Most authors register and classify adverse events using the Common Terminology Criteria for Adverse Events version 5.0 (CTCAE v5.0) [2,16,17,23], which is the gold standard in cancer clinical trials, while others use the Clavien–Dindo scale [13,20,27,28], as we did in our study. The issue of heterogeneity in complication reporting has recently been addressed by the PIPAC UK collaborative group in a systematic review. They proposed that CTCAE should be the standard reporting measure, along with 30-day postoperative mortality, in future prospective trials due to its comprehensive assessment, especially when PIPAC is delivered together with systemic chemotherapy [14].

Another important aspect of novel cancer treatment is its tolerability. In the present study, 71.9% of patients received more than two PIPAC procedures, and 43.8% received more than three PIPAC procedures. There were no reported failures for laparoscopic abdominal access, indicating that the discontinuation of PIPAC was due to the deterioration of general health rather than technical reasons. A higher number of PIPAC cycles has been reported to be associated with improved survival [2,16,19,20,29]. However, it remains unclear whether healthier patients survive longer and can tolerate more PIPAC cycles or if receiving more PIPAC cycles directly prolongs survival. In our cohort, the median OS after the diagnosis of PM was 12.5 months, and the median survival after the first PIPAC procedure was 5 months. These figures align with previous reports, which show a median OS ranging from 8 to 19.1 months [13,30] and survival after the first PIPAC ranging from 4.7 to 6.9 months [2,14,16,19,20,29]. Such relatively short survival after initiating PIPAC treatment must be considered carefully. In most previous studies, as well as in our cohort, the vast majority of patients received PIPAC late in the treatment pathway after the failure of several lines of systemic chemotherapy [14]. Administering PIPAC earlier, before the development of chemoresistance to systemic treatment, may increase the proportion of patients able to receive more PIPAC cycles and potentially improve treatment efficacy. Moreover, implementing PIPAC in the early stages of treatment may allow for its use in a bidirectional manner when combined with systemic chemotherapy. Although there is no clear evidence to date showing whether bidirectional therapy adds additional benefits, it has the potential to optimize both systemic and local (peritoneal) control [14].

In general, the present study suggests that PIPAC may be a valuable treatment option for selected patients with GC PM, offering a low rate of treatment-related complications and potentially promising survival outcomes.

However, several limitations must be considered when interpreting these results. The retrospective nature of the study and the relatively small sample size are significant constraints, as they may affect the robustness and generalizability of the findings. Additionally, the absence of a control group receiving standard care without PIPAC limits our ability to draw definitive conclusions about the comparative efficacy of PIPAC versus other treatment modalities. To address these limitations and provide a more comprehensive understanding of PIPAC’s role in the treatment of GC PM, continued research is essential. Larger prospective studies are warranted to further elucidate PIPAC’s effectiveness, optimize treatment protocols, and identify patient populations who may benefit the most from this innovative approach. Such research will help to clarify PIPAC’s role within the broader context of treatment for GC PM and contribute to the development of more effective management strategies for this challenging clinical condition.

## 5. Conclusions

The present study suggests that PIPAC is a feasible and safe treatment option for patients with GC PM. Despite the non-significant reductions in PCI and ascites volume, PIPAC’s potential to stabilize the disease and its acceptable safety profile underscores its utility as part of a multimodal treatment strategy. Continued research, including larger and prospective studies, is warranted to further elucidate the benefits and optimize the use of PIPAC in this challenging clinical context.

## Figures and Tables

**Figure 1 cancers-16-02992-f001:**
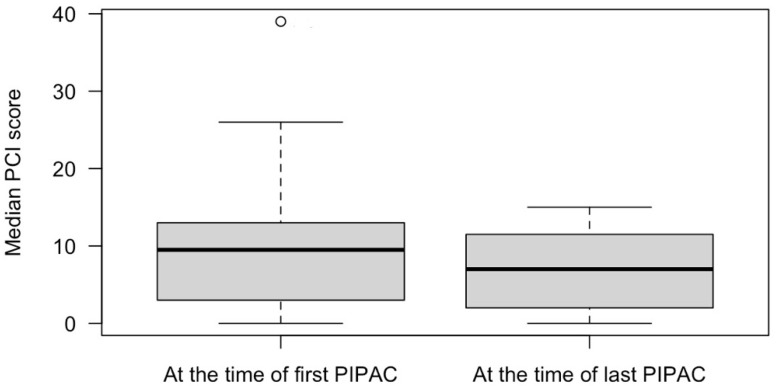
Median peritoneal carcinomatosis index in patients who received PIPAC procedures for gastric cancer peritoneal metastases.

**Figure 2 cancers-16-02992-f002:**
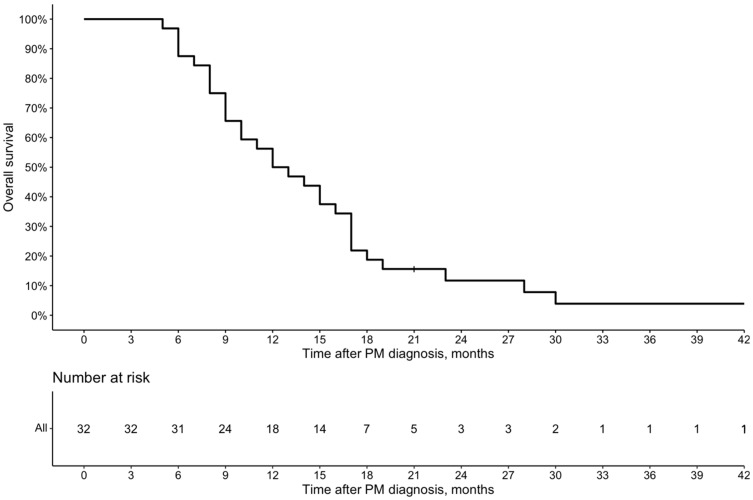
Overall survival from the PM diagnosis of patients who received PIPAC peritoneal metastases.

**Figure 3 cancers-16-02992-f003:**
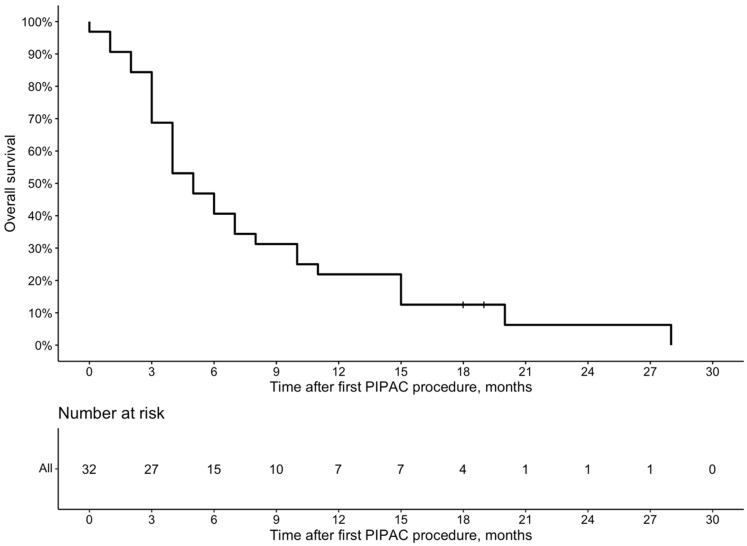
Overall survival from the first PIPAC procedure due to GC peritoneal metastases.

**Table 1 cancers-16-02992-t001:** Baseline clinicopathologic characteristics of patients who received PIPAC.

Sex, n (%)	Female	15 (46.9%)
	Male	17 (53.1%)
Median age (Q1; Q3), years		55 (46; 66)
Median hospitalization (Q1; Q3), days		2 (1; 4.3)
Median BMI (Q1; Q3)		22.7 (20.3; 25.1)
History of radical surgery for primary tumor, n (%)	Yes	5 (15.6%)
	No	27 (84.4%)
Median CEA level (Q1; Q3) at the time of first PIPAC, ng/L		1.9 (0.98; 5)
Median CA19.9 level (Q1; Q3) at the time of first PIPAC, ng/L		12.8 (3.6; 77.1)
Number of PIPAC procedures, n (%)	1	9 (28.1%)
	2	9 (28.1%)
	3	13 (40.6%)
	5	1 (3.1%)
Median operation time (Q1; Q3), min		92.5 (85; 110)

## Data Availability

The data presented in this study are available on reasonable request from the corresponding author due to local legal regulations.

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
