# Peer review of "Pressurized Intraperitoneal Aerosol Chemotherapy (PIPAC) for Gastric Cancer Peritoneal Metastases: Results from the Lithuanian PIPAC Program"

_cancers, 2024, doi:10.3390/cancers16172992_

Round 1

Reviewer 1 Report

Comments and Suggestions for Authors

This is a clearly written paper and the conclusion and discussion are balanced . The outcomes are consistent with the literature of similar studies.  The study population here generally did not have a high PCI score with a median of 10 at baseline.  It would be helpful if the authors included progression free survival.  I infer from the discussion, that patients did not receive intervening systemic chemo between PIPAC which is being done at some institutions, and would be grateful if the author could make an comment on this in the methods.

The weakness of the study  as the authors have identified is the lack of CTCAE toxicity grading and PRGS reporting.

Author Response

This is a clearly written paper and the conclusion and discussion are balanced. The outcomes are consistent with the literature of similar studies.  The study population here generally did not have a high PCI score with a median of 10 at baseline.  It would be helpful if the authors included progression free survival.  I infer from the discussion, that patients did not receive intervening systemic chemo between PIPAC which is being done at some institutions, and would be grateful if the author could make an comment on this in the methods.

The weakness of the study  as the authors have identified is the lack of CTCAE toxicity grading and PRGS reporting.

We would like to express our sincere gratitude to the reviewer for his/her insightful and positive feedback on our paper. Unfortunately, progression-free survival (PFS) data was not available for the specific cohort analyzed in this study, which is why it was not included in the manuscript. We appreciate the reviewer for highlighting the lack of clarity regarding the application of PIPAC as a bidirectional treatment. We have addressed this in the methods section with the following clarification: “The decision to administer PIPAC bidirectionally, with systemic chemotherapy applied between PIPAC procedures, or as a unimodal treatment option, was made on a case-by-case basis during multidisciplinary team meetings.” Additionally, we have clarified this in the results section: “Twenty patients (62.5%) received PIPAC as a bidirectional treatment in combination with systemic therapy, while 12 patients (37.5%) received it as a unimodal treatment.”

Reviewer 2 Report

Comments and Suggestions for Authors

Martynas Luksta and co-authors present a comprehensive and well-written manuscript focused on the outcomes of pressurized intraperitoneal aerosol chemotherapy (PIPAC) for gastric cancer peritoneal metastases (GC PM) within the Lithuanian PIPAC program. 

The authors aimed to summarize the outcomes of GC PM treatment with PIPAC from the Lithuanian PIPAC program. For that they investigated the safety and efficacy of PIPAC, providing detailed data from a retrospective cohort study conducted at Vilnius University Hospital Santaros Klinikos.

Authors included the patients who underwent PIPAC for GC PM at Vilnius University Hospital Santaros Klinikos between 2015 and 2022 were included in this retrospective study. The safety of PIPAC was assessed by postoperative complications according to the Clavien-Dindo classification. Efficacy was evaluated based on the peritoneal carcinomatosis index (PCI), ascites dynamics throughout the treatment, and long-term outcomes.

Authors begin with an in-depth background on the challenges of treating GC PM and the limitations of current systemic therapies. They then transitions into a meticulous description of the PIPAC procedure, patient selection, and data collection methods. 

The study includes 32 patients who underwent a total of 71 PIPAC procedures, with a focus on perioperative outcomes, changes in the peritoneal carcinomatosis index (PCI), and ascites volume.

The authors present clear and well-supported results, indicating that PIPAC is a safe and feasible treatment option for GC PM, despite the lack of statistically significant reductions in PCI and ascites volume. The discussion contextualizes these findings within the broader literature, acknowledging the limitations of the study, including its retrospective design and small sample size, while emphasizing the need for further prospective research.

Finally, authors conclude that PIPAC is a safe and feasible treatment option for GC PM,  however, well-designed prospective studies are needed to fully assess its efficacy.

Overall, the manuscript provides valuable insights into the potential benefits and limitations of PIPAC as a treatment for GC PM and is a significant contribution to the field.

======================

Other comments to authors:

1) Please check for typos throughout the manuscript.

2) Please improve figures/tables where appropriate.

3) With regards to drug delivery - authors are kindly encouraged to cite the following article that describes drug delivery approaches relevant for gastric cancer peritoneal metastases. DOI: 10.1016/j.nantod.2023.102004

Author Response

Martynas Luksta and co-authors present a comprehensive and well-written manuscript focused on the outcomes of pressurized intraperitoneal aerosol chemotherapy (PIPAC) for gastric cancer peritoneal metastases (GC PM) within the Lithuanian PIPAC program.

The authors aimed to summarize the outcomes of GC PM treatment with PIPAC from the Lithuanian PIPAC program. For that they investigated the safety and efficacy of PIPAC, providing detailed data from a retrospective cohort study conducted at Vilnius University Hospital Santaros Klinikos.

Authors included the patients who underwent PIPAC for GC PM at Vilnius University Hospital Santaros Klinikos between 2015 and 2022 were included in this retrospective study. The safety of PIPAC was assessed by postoperative complications according to the Clavien-Dindo classification. Efficacy was evaluated based on the peritoneal carcinomatosis index (PCI), ascites dynamics throughout the treatment, and long-term outcomes.

Authors begin with an in-depth background on the challenges of treating GC PM and the limitations of current systemic therapies. They then transitions into a meticulous description of the PIPAC procedure, patient selection, and data collection methods.

The study includes 32 patients who underwent a total of 71 PIPAC procedures, with a focus on perioperative outcomes, changes in the peritoneal carcinomatosis index (PCI), and ascites volume.

The authors present clear and well-supported results, indicating that PIPAC is a safe and feasible treatment option for GC PM, despite the lack of statistically significant reductions in PCI and ascites volume. The discussion contextualizes these findings within the broader literature, acknowledging the limitations of the study, including its retrospective design and small sample size, while emphasizing the need for further prospective research.

Finally, authors conclude that PIPAC is a safe and feasible treatment option for GC PM,  however, well-designed prospective studies are needed to fully assess its efficacy.

Overall, the manuscript provides valuable insights into the potential benefits and limitations of PIPAC as a treatment for GC PM and is a significant contribution to the field

Other comments to authors:

1) Please check for typos throughout the manuscript.

2) Please improve figures/tables where appropriate.

3) With regards to drug delivery - authors are kindly encouraged to cite the following article that describes drug delivery approaches relevant for gastric cancer peritoneal metastases. DOI: 10.1016/j.nantod.2023.102004

We would like to thank the reviewer for taking the time and effort to critically evaluate our manuscript. As suggested, we have checked for typos and ensured the quality of the figures and tables. We also agree with the reviewer that, in addition to IP chemotherapy, various other methods are currently being developed to treat PM. Accordingly, we have cited the recommended manuscript.

Reviewer 3 Report

Comments and Suggestions for Authors

This retrospective analysis of 32 patients is very interesting as metastatic gastric cancer is incurable and peritoneal metastses may cause severe problem due to ascites. The PIPAC programme needs very careful patient selection. 

my questions: why the german first study did not continue, and PIPAC is available just in Lithuania ?

why 6 weekly do you repeat surgery? I think it is very uncomfortable to the patients (and dangerous to enhance the possibility of thromboembolic events..) and time by time the intraperitoneal adhesions may cause complication.

why do you choose gastric cancer and compare with HIPEc that is suitable for mucinotic tumors of appendiy and ovarian cancer.

Following perform minor revision I accept the manuscript.

Comments on the Quality of English Language

Following perform minor revision I accept the manuscript.

Author Response

This retrospective analysis of 32 patients is very interesting as metastatic gastric cancer is incurable and peritoneal metastses may cause severe problem due to ascites. The PIPAC programme needs very careful patient selection.

my questions: why the german first study did not continue, and PIPAC is available just in Lithuania ?

why 6 weekly do you repeat surgery? I think it is very uncomfortable to the patients (and dangerous to enhance the possibility of thromboembolic events..) and time by time the intraperitoneal adhesions may cause complication.

why do you choose gastric cancer and compare with HIPEc that is suitable for mucinotic tumors of appendiy and ovarian cancer.

All authors of the manuscript would like to express our sincere gratitude to the reviewer for his/her valuable comments on our paper.

First, we would like to clarify that PIPAC is currently under investigation and experimental clinical application in various countries, not only in Lithuania. To the best of our knowledge, several centers in Germany are also investigating PIPAC for PM.

We fully agree with the reviewer that repeated surgeries required for PIPAC applications may pose a risk to patients. However, as our results have shown and as discussed in the manuscript, minimally invasive surgeries are relatively safe for patients, and repeated surgeries for drug applications offer the opportunity to objectively assess tumor responses by comparing biopsies obtained at different stages of treatment.

We also concur that GC PM is not an established indication for HIPEC. Nonetheless, we have acknowledged the limitations of the current knowledge in the introduction.